# Build Deep Neural Network Models to Detect Common Edible Nuts from Photos and Estimate Nutrient Portfolio

**DOI:** 10.3390/nu16091294

**Published:** 2024-04-26

**Authors:** Ruopeng An, Joshua M. Perez-Cruet, Xi Wang, Yuyi Yang

**Affiliations:** 1Brown School, Washington University in St. Louis, St. Louis, MO 63130, USA; w.xi@wustl.edu; 2School of Medicine, Washington University in St. Louis, St. Louis, MO 63130, USA; j.m.perez-cruet@wustl.edu; 3Division of Computational and Data Science, Washington University in St. Louis, St. Louis, MO 63130, USA

**Keywords:** neural network, deep learning, nut, diet, computer vision, object detection

## Abstract

Nuts are nutrient-dense foods and can be incorporated into a healthy diet. Artificial intelligence-powered diet-tracking apps may promote nut consumption by providing real-time, accurate nutrition information but depend on data and model availability. Our team developed a dataset comprising 1380 photographs, each in RGB color format and with a resolution of 4032 × 3024 pixels. These images feature 11 types of nuts that are commonly consumed. Each photo includes three nut types; each type consists of 2–4 nuts, so 6–9 nuts are in each image. Rectangular bounding boxes were drawn using a visual geometry group (VGG) image annotator to facilitate the identification of each nut, delineating their locations within the images. This approach renders the dataset an excellent resource for training models capable of multi-label classification and object detection, as it was meticulously divided into training, validation, and test subsets. Utilizing transfer learning in Python with the IceVision framework, deep neural network models were adeptly trained to recognize and pinpoint the nuts depicted in the photographs. The ultimate model exhibited a mean average precision of 0.7596 in identifying various nut types within the validation subset and demonstrated a 97.9% accuracy rate in determining the number and kinds of nuts present in the test subset. By integrating specific nutritional data for each type of nut, the model can precisely (with error margins ranging from 0.8 to 2.6%) calculate the combined nutritional content—encompassing total energy, proteins, carbohydrates, fats (total and saturated), fiber, vitamin E, and essential minerals like magnesium, phosphorus, copper, manganese, and selenium—of the nuts shown in a photograph. Both the dataset and the model have been made publicly available to foster data exchange and the spread of knowledge. Our research underscores the potential of leveraging photographs for automated nut calorie and nutritional content estimation, paving the way for the creation of dietary tracking applications that offer real-time, precise nutritional insights to encourage nut consumption.

## 1. Introduction

A nut is a fruit consisting of a hard or tough shell protecting an edible kernel [1]. Both tree nuts and peanuts are rich in unsaturated fat and other essential nutrients, such as vitamins, minerals, fibers, and high-quality vegetable proteins [2]. Many nuts, such as almonds, walnuts, and cashews, are consumed as snacks and serve as popular ingredients [3]. Nut consumption contributes to a healthy, nutritious diet, as recommended in the 2020–2025 Dietary Guidelines for Americans [4]. Previous studies have shown an association between nut consumption and reduced incidence of diabetes and coronary heart disease [5,6]. Potential benefits of nut consumption also include improvements in hypertension, visceral obesity, hyperlipidemia, inflammation, oxidative stress, and metabolic syndrome [1,7].

Precise, immediate nutritional data can guide decisions related to diet [8]. Many nuts of one or multiple types (commonly called “mixed nuts”) are often contained in a packet where the nutrition facts label is printed. The label provides per-serving nutrient portfolios (e.g., protein, carbohydrate, total fat, and saturated fat) for each nut type separately or mixed types as a whole. This way of presentation impedes nut consumers’ access and assessment of relevant nutritional information: the nutrition facts label may not be readily available to a consumer during consumption; and even given its accessibility, it is cognitively tasking to calculate the overall nutrient intakes from a handful of nuts of the same or different types.

Smartphone diet-tracking apps are effective in monitoring daily eating patterns, weight changes, and chronic conditions [9,10]. A recent literature review indicates that these applications are highly user-friendly and accurately track daily calorie and nutrient consumption [9]. They also demonstrated a positive impact on dietary behavior [11]. The primary technology underpinning the majority of diet-tracking applications is deep learning [12]. Deep learning relies on artificial neural networks, employing several layers of processing to incrementally extract more complex features from the data [13]. The hierarchical structure of this representation facilitates the identification of intricate patterns, proving particularly useful for analyzing large datasets, including images, videos, audio, and text [14]. Recent studies suggest that applications of deep learning for image classification and object detection could enhance dietary assessment by increasing efficiency and reducing inaccuracies associated with self-reported food intake [15,16]. The rapid increase in diet-tracking apps [9,10] indicates a growing demand from nutrition professionals and the public for automated diet monitoring. This demand should be met by data acquisition and model building.

Data accessibility and quality play a central role in designing and implementing artificial intelligence (AI) models [17]. Datasets of food images, like Fruits 360, which includes around 90,000 images of 131 types of fruits and vegetables, and Food-101, with around 100,000 images of 101 varieties of food, have been created for these objectives. Yet, there is still a scarcity of systematically gathered and labeled image data related to frequently eaten nuts. Dheir et al. used deep learning to classify five nut types (i.e., peanut, pecan, almond, chestnut, and hazelnut) [18], but the relevant dataset was publicly unavailable. Our team developed and publicly released a dataset comprising 2200 images showcasing 11 varieties of nuts [19]. We fine-tuned neural network models to classify nut types with high accuracy based on the dataset. However, three gaps remain. First, the dataset contained one nut per image, whereas, in real-world situations, people usually consume multiple nuts of one or mixed types on a single occasion. Second, each image in the dataset is labeled by the nut type it contains, but the exact location of the nut in the image is unannotated, making the dataset suitable for nut classification tasks but unusable for nut detection and localization tasks. Finally, the model developed based on the dataset can only classify nut types if a single nut (type) appears in an image, making it incapable of estimating the nutrient portfolio of a handful of mixed nuts.

This study contributes to the scientific literature by addressing the above gaps in our previous work. We hypothesize that by using transfer learning and deep neural network models, we can accurately identify and locate various nuts in an image, which in turn allows for the estimation of their combined nutritional profiles. To support this, we created a dataset with 1380 images featuring 11 types of nuts that are frequently consumed. An image includes three nut types; each type consists of two to four nuts, so six to nine nuts are in each image. Every nut in the dataset was marked with a rectangular bounding box to indicate its location within the image, rendering the dataset ideal for training models capable of multi-label classification and object detection. Using transfer learning, deep neural network models were developed to identify and pinpoint the positions of nuts in an image. By combining nut-type-specific nutritional information, the model can accurately estimate the aggregate nutrient portfolios (e.g., total energy, protein, carbohydrate, total fat, and saturated fat) of all nuts captured in a photo. The study may spur the deployment of AI models to inform nut consumption as part of a nutritious diet.

## 2. Methods

### 2.1. Data

We chose 11 commonly consumed nut types—almond, Brazil nut, cashew, chestnut, hazelnut, macadamia, peanut, pecan, pine nut, pistachio, and walnut. We purchased shelled nuts from online or local stores. We used an iPhone 11 to take photos of nut samples because diet-tracking apps commonly use mobile phones. Each image was captured in the JPEG format with a resolution of 4032 × 3024 pixels. An image included six to nine nuts in total, belonging to three types. All nuts were displayed randomly.

Initially, the images were resized to 512 × 512 pixels, employing different cropping techniques: random cropping for the training set and center cropping for the validation set. After undergoing data augmentation, the images were further downscaled to 384 × 384 pixels prior to input into the models. This method of progressive resizing, implemented by fastai2, is used to enhance model performance.

The 11 nut types are equally represented in the dataset of 1380 images. The VGG Image Annotator (VIA) Version 2 was used to label all the images in the dataset [20], where each nut was manually encircled with a rectangular bounding box using VIA and its type was specified. The dataset was then randomly split into three parts: the training set comprising 80% of the total (1104 images), the validation set making up 10% (138 images), and the test set constituting the remaining 10% (138 images).

### 2.2. Model

Neural network models utilized the IceVision library [21] to learn how to identify nuts and their locations (represented by rectangular bounding boxes) within images. IceVision serves as a comprehensive computer vision framework, providing access to a selection of top-performing pre-trained models sourced from libraries like Torchvision, MMDetection, YOLO, and EfficientDet. It facilitates a seamless deep learning process, from start to finish, by integrating powerful libraries such as PyTorch Lightning and fastai2 for model training.

Transfer learning enables the application of insights gained from addressing one problem to a different, yet related, problem [22]. For instance, the understanding developed during the process of recognizing oranges, encapsulated in the trainable parameters of neural networks, can be leveraged to educate another model on identifying apples. We employed transfer learning by utilizing four pre-trained models: Faster R-CNN, RetinaNet, YOLOv5, and EfficientDet, each having unique architectures and being extensively used for object detection tasks. Faster R-CNN is a model that combines region proposal networks and fast R-CNN by sharing their convolutional features [23]. RetinaNet is known for its focal loss function, which addresses class imbalance during training [24]. YOLOv5 is an iteration of the You Only Look Once (YOLO) model, optimized for speed and accuracy with a single neural network [25]. EfficientDet integrates efficient scaling methods and a compound coefficient to balance network depth, width, and resolution, providing a scalable and efficient object detector [26].

To improve model performance, techniques such as data augmentation, normalization, and optimization of the learning rate were employed. Data augmentation enriches the training dataset’s variety through the application of random yet plausible transformations, aiding in the prevention of model overfitting [27]. Before being inputted into the model, images in the training set underwent a series of data augmentation procedures such as resizing, zooming, cropping, rotating, and adjusting the contrast. Data normalization facilitates model convergence by ensuring trainable parameters share similar statistical distributions. Learning rate (LR) is a critical hyperparameter in model training. A learning rate (LR) that is set too low can lead to extended training durations and potentially suboptimal results, while an LR set too high risks surpassing the optimal point for the model. The LR finder tool in fastai2 employs a cyclical learning rate strategy, enabling the LR to fluctuate within sensible limits during training [28]. The LR recommendations provided by the LR finder were utilized for both the pre-freezing and post-freezing stages of model training.

Our approach to training the model involved a two-phase strategy. Initially, during the pre-freezing phase, we trained the object detection model’s head—which is responsible for identifying and classifying nuts in an image—with randomly initialized weights, on top of a base model whose trainable weights were kept frozen. Following this, in the post-freezing phase, we unfroze all the weights of the model and trained them simultaneously. Due to the necessity of data augmentation, more epochs were required for model training, leading to 10 epochs in the pre-freezing phase and 20 epochs in the post-freezing phase. The model from the pre-freezing phase that exhibited the lowest validation loss was selected for further training in the post-freezing phase. Similarly, the post-freezing model demonstrating the lowest validation loss was chosen as the final model. This final model was then evaluated on the test set by comparing its predictions against the actual labels. The evaluation metric used was the mean average precision (mAP), a standard measure in object detection that quantifies the model’s accuracy. It calculates the average precision (AP) for each class and takes the mean of these values. The AP for each class is computed as the area under the precision–recall curve, reflecting both the precision (proportion of true positive predictions among all positive predictions) and the recall (proportion of true positive predictions among all actual positives) of the model [29]. The mAP score ranges from zero to one, where a higher score indicates higher accuracy and precision in detecting and correctly labeling objects within the images [29]. For nutritional analysis, we used the USDA National Nutrient Database for Standard Reference to obtain accurate nutritional values of the nuts [30]. The entire modeling process was conducted using Python 3.10, and model training was expedited by a Tesla V100 GPU (NVIDIA, Santa Clara, CA, USA).

## 3. Results

We developed and labeled a dataset comprising 1380 images, featuring 11 varieties of nuts: almond, Brazil nut, cashew, chestnut, hazelnut, macadamia, peanut, pecan, pine nut, pistachio, and walnut. Each image includes three nut types, with six to nine nuts in total. Figure 1 shows some sample images. In each image, every nut is encased within a rectangular bounding box, which is marked with the nut’s type.

The dataset was randomly divided into training (1104 images), validation (138 images), and test sets (138 images). We fine-tuned four pre-trained models—Faster R-CNN, RetinaNet, YOLOv5, and EfficientDet—by training them on our dataset and then assessed their effectiveness on the validation set using the mean average precision (mAP) metric. Table 1 reports the evaluation results, documenting the highest mAP score and the corresponding number of epochs. The YOLOv5 model outperformed the others, achieving the highest mean average precision (mAP) score of 0.7596. It was followed by EfficientDet with a score of 0.7250, Faster R-CNN at 0.7114, and RetinaNet trailing at 0.4905. Based on these results, YOLOv5 was selected as the definitive model.

Figure 2 displays comparative examples of the actual annotations, including bounding boxes and labels, alongside the predictions made by YOLOv5 on the test set. The predicted bounding boxes closely matched the actual locations and dimensions, and the labels were accurately forecasted with high confidence.

The primary objective of this study was to ascertain the quantity and types of nuts in an image, enabling the calculation of comprehensive nutrient profiles, such as total calories and protein content, rather than pinpointing the exact location of each nut (i.e., bounding box). Therefore, we evaluated the number and proportion of matches between actual annotations and labels predicted by YOLOv5 in the test set. To facilitate this, we constructed two matrices: one representing the ground truth and the other reflecting the predicted labels. In both matrices, each row corresponds to an individual image from the test set, while the 11 columns represent the different types of nuts. The value in each cell of the ground truth (predicted) matrix indicates the actual (predicted) count of a particular nut type in an image. We then compared the values in corresponding cells across the two matrices. Across 1518 cell-level comparisons (138 images × 11 nut types per image), a match was identified when two corresponding cells had identical values. Out of 1518 comparisons, there were 1486 matches, resulting in a prediction accuracy of 97.9% for the YOLOv5 model.

To derive comprehensive nutrient profiles for each image, we integrated specific nutritional data for each type of nut, such as calories per serving, sourced from the US Department of Agriculture’s FoodData Central, with the YOLOv5 model’s predictions (i.e., the identified types and quantities of nuts in an image). This approach allowed us to estimate 12 aggregate nutrient portfolios for all the nuts depicted in an image: total energy, protein, carbohydrate, total fat, saturated fat, fiber, vitamin E, magnesium, phosphorus, copper, manganese, and selenium. Table 2 outlines the discrepancy rates between the nutrient estimates based on YOLOv5′s predictions and the actual (ground truth) nutrient content. The discrepancies across these 12 nutrients varied, with the lowest being 0.8% for selenium and the highest reaching 2.6% for carbohydrates.

## 4. Discussion

Nuts are nutrient-rich foods with documented cardiovascular and metabolic benefits and can be integrated into a healthy eating pattern [1]. AI-powered diet-tracking apps could advise nut consumption depending on data and model availability. We created a dataset consisting of 1,380 images featuring 11 widely consumed types of nuts, with each nut in an image marked by a rectangular bounding box. Leveraging this dataset, we trained neural network models to recognize and pinpoint the locations of nuts. The refined model is capable of accurately calculating the nutritional content of various nuts within an image, showcasing the potential for automated nut calorie and nutrition tracking through photography. To promote data sharing and the spread of knowledge, we have made our datasets and models publicly available.

Deep neural network models typically need large datasets for training. Model training is costly, generating a large carbon footprint [31]. However, data augmentation, transfer learning, and learning rate optimization have substantially improved model performances on datasets with a small or moderate sample size. The final model is lightweight, with 27.1 MB. Embedding the model in a mobile phone app may enable users to estimate the energy and nutritional intake from nuts in real time.

Images of nuts gathered from this and future projects can be combined with existing food image databases like Food-101, enabling the training of neural network models to recognize a broader variety of food items. These applications are suitable for real-time diet monitoring, which reduces the cognitive burden of those who track daily energy and nutrient intakes for weight or disease management [9,10]. Dietitians and nutritionists may also design interventions using data collected by diet-tracking apps.

People’s dietary behavior is influenced by many psychosocial factors [32,33]. Accurate, real-time measures for dietary intake alone are insufficient to motivate behavioral changes [34,35,36]. Besides constructing state-of-the-art AI models, diet-tracking app developers should consider the applications of theories to guide psychological and behavioral modifications [37]. In particular, they may pay attention to improving knowledge, motivation, attitudes, self-efficacy, and goal setting to facilitate users’ transition to a healthier diet [11].

This study has several limitations. The size of the dataset is modest and primarily encompasses whole, easily recognizable nuts, neglecting the variability introduced by broken nuts or nut pieces. Additionally, our dataset does not include other common ingredients found in nut mixtures, such as raisins or goji berries, which may affect the model’s performance in real-world scenarios. Furthermore, we observed a higher error rate in carbohydrate estimations compared to other nutrients. This discrepancy can be attributed in part to the larger numeric values associated with carbohydrates, which amplify errors in cases of misclassification or incorrect nut identification. These challenges highlight the need for more comprehensive data and refined modeling techniques to improve the accuracy of nutritional estimations in diverse real-world applications. Addressing these aspects will enhance the generalizability and applicability of the AI models. Moreover, there is a potential risk that the app’s recommendations could lead to overconsumption. To address this concern, it is advisable to provide balanced nutritional information that emphasizes both the health benefits and the recommended consumption limits. Additionally, the development of monitoring and feedback mechanisms, which would alert users when their intake approaches or exceeds these limits, is essential.

This study serves as a proof-of-concept experiment, showcasing the practicality and benefits of training neural network models to identify nuts in photographs. Nevertheless, it leaves a lot of questions open for further investigation. How can we reduce the model size without compromising performance so the model can be deployed to low-end mobile devices with limited storage? Is the model capable of accurately recognizing nuts in user-provided photos that vary from the images in its training dataset? Those without smartphones who want to use diet-tracking apps could upload photos to the cloud for model inference. How can we protect the confidentiality of those users’ data? Image input represents just one way in which individuals can engage with AI systems; other common methods of interaction include textual, speech, and video inputs and data from sensors [38]. Employing multimodal inputs for human–AI interactions has the potential to boost model performance and dependability [39]. However, the application of such approaches in dietary interventions is still in its early stages and remains relatively underexplored. Our project used about 30 samples for each nut type, which may be insufficient for generalizability. We will add more samples to the database to bolster data diversity in future endeavors.

## 5. Conclusions

In this study, we developed and labeled a dataset with 1380 images showcasing 11 types of nuts that are widely consumed. Using this dataset, we trained neural network models to recognize various nut types. The nutritional content estimated by our final model closely matched the actual data, demonstrating the model’s accuracy in identifying and analyzing the nutrient profiles of nuts. Our findings highlight the potential of AI-powered diet-tracking applications to provide real-time, accurate nutritional information, thereby supporting healthier eating habits and diet quality improvement. This study may shed light on innovative behavioral interventions influencing nut consumption by providing consumers with instantaneous, precise nutritional information. The models and app may also help researchers and practitioners measure and collect individuals’ nut consumption data in real time, addressing the recall error and social desirability bias often contaminating food frequency questionnaires and 24 h dietary interviews.

## Figures and Tables

**Figure 1 nutrients-16-01294-f001:**
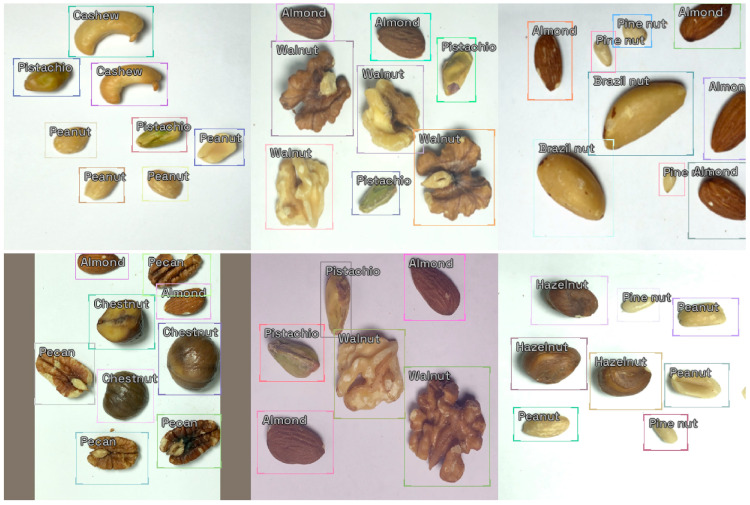
Sample images from the nut type dataset with annotated bounding boxes.

**Figure 2 nutrients-16-01294-f002:**
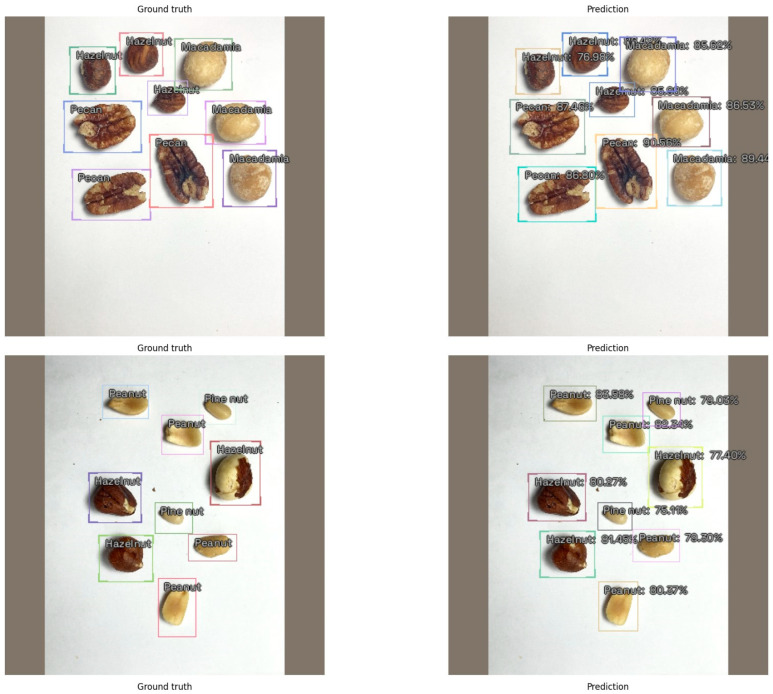
Comparison of ground truth annotations and YOLOv5 predictions for nut detection in the test set.

**Table 1 nutrients-16-01294-t001:** Model performances on validation set measured by mean average precision (mAP).

Model	Highest mAP Score	Number of Epochs
Faster R-CNN	0.7114	17
RetinaNet	0.4905	19
YOLOv5	0.7596	19
EfficientDet	0.7250	13

**Table 2 nutrients-16-01294-t002:** Proportions of discrepancies between model-predicted and ground truth image-level aggregate nutrient portfolios.

Model	Predicted Mean	Ground Truth Mean	Proportion of Discrepancy ± Standard Error
Total energy (kcal)	101.24	99.68	1.56%
Protein (g)	2.36	2.33	1.13%
Carbohydrate (g)	4.32	4.21	2.58%
Total fat (g)	9.02	8.90	1.37%
Saturated fat (g)	1.42	1.40	1.18%
Fiber (g)	1.14	1.13	1.34%
Vitamin E (mg)	0.78	0.77	1.31%
Magnesium (mg)	36.67	36.20	1.28%
Phosphorus (mg)	71.59	70.86	1.02%
Copper (mg)	0.20	0.20	1.19%
Manganese (mg)	0.42	0.41	2.19%
Selenium (µg)	85.76	85.06	0.82%

## Data Availability

Data are contained within the article.

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
