# Peer review of "Build Deep Neural Network Models to Detect Common Edible Nuts from Photos and Estimate Nutrient Portfolio"

_nutrients, 2024, doi:10.3390/nu16091294_

Round 1
Reviewer 1 Report
Comments and Suggestions for Authors
The study concerns in building a model able to recognize different types of edible nuts from photos in order to analyze their nutritional value. It is very interesting new type of study probably worth publishing but it needs to be corrected. This study lacks of typical structure, authors should indicate the aim of the study, what was the hypothesis. Consequently, you should refer to your aim and hypothesis in conclusions.
You presented on the pictures nicely shaped nuts, easy to recognize. What about broken nuts or pieces of nuts? What about other ingredients which are also present in such mixtures, like raisins or goji berries? Does your model recognize them? If you did not do such experiments, you should indicate it in limitations at least.
Conclusions is not a discussion, so you should not apply references there but only conclusions drawn from your study. Your conclusions is rather summary, what can be concluded from your study?
Errors need to be corrected:
Line 26 - vitamin E is not a mineral
Line 74 – it should be Dheir et al.
What kind of database you applied to analyze nutritional value of the analyzed nuts (table 2)? What software you applied to analyze discrepency ± standard error? You should describe it in methods.
Reviewer 2 Report
Comments and Suggestions for Authors
Article:
Build Deep Neural Network Models to Detect Common Edible Nuts from Photos and Estimate Nutrient Portfolio
Study describes the generation of a dataset of images of multiple nut types. Then training neural net models to correctly identify nuts in the image datasets. Study was conducted to test the accuracy of possible future diet-tracking software applications (i.e. smart phones) to identify nuts consumed by users, and to provide accurate nutrient information.
Questions/Suggestions
Line 15: "To facilitate the identification of each nut, rectangular bounding boxes were employed to delineate their locations within the images. "
Please provide more information in abstract. ie. software/programming language, image type/size
Line 19: "Utilizing transfer learning, deep neural network models were adeptly trained to recognize and pinpoint the nuts depicted in the photographs"
As above, please provide a little more information in abstract. i.e. state software/programming language
Line 21: "a mean average precision of 0.7596"
Question: What does this mean, how does this differ from "97.9% accuracy rate"?
Line 89: "Every nut in the dataset was marked with a rectangular bounding box to indicate its location within the image"
Question: are unmarked images also made available?
Line 101: "We purchased shelled nuts from online or local store"
Question: raw or roasted?
Line 102: " We used an iPhone 11 to take photos of nut samples because diet-tracking apps commonly use mobile phones"
Question: What was image format, image size?
Line 106: "where each nut was encircled with a rectangular bounding box".
If this was done manually, please change above to "where each nut was manually encircled with a rectangular bounding box using VGG image annotator"
Question: can bounding boxes be applied automatically? wouldn’t this be preferred from an app perspective?
Line 107: "and its type was specified"
Question: what does this mean, where was this data entered?
Line 112: "IceVision library"
citation(s) needed
Line 126: "The photographs captured with an iPhone featured dimensions of..."
This information (paragraph) should be moved to "Data" section.
Line 130: "images were further downscaled to 384 × 384 pixels"
Question: 384x384 pixel images were used to view 6-9 nuts? What was the average size of each nut? (in pixels), why downsize images to this extent? Wouldn't higher pixel resolutions give more accurate results?
Line 133: "Data augmentation enriches the training dataset's variety through the application of random yet plausible transformations, aiding in the prevention of model overfitting."
citation(s) needed
Line 135: "Before being inputted into the model, images in the training set underwent a series of data augmentation procedures such as resizing, zooming, cropping, rotating, and adjusting the contrast."
Question: How is this "augmentation" not divorcing the images from reality? i.e. how is this not artificially providing an idealized data set that will never be encountered in real life? Rendering training set not very useful.
Line 168: Figure 1.
Comment: Some nuts are very commonly found as half’s (e.g. Walnuts), were Walnuts half’s also imaged for identification? What about other common nut fragmentations?
Line 171: "four pre-trained models—Faster R-CNN, RetinaNet, YOLOv5, and EfficientDet"
Comment: Please better outline what these are in your methods section
Line 176: "mean average precision (mAP) score"
Comment: Please better define in text what this number means?
Round 2
Reviewer 1 Report
Comments and Suggestions for Authors
I accept the revised version
Author Response
Thank you very much.